# Skilful decadal-scale prediction of fish habitat and distribution shifts

Mark R. Payne [1,2 ✉], Gokhan Danabasoglu[3], Noel Keenlyside [4,5], Daniela Matei[6], Anna K. Miesner [7], Shuting Yang [1] & Stephen G. Yeager [3]

Many fish and marine organisms are responding to our planet's changing climate by shifting their distribution. Such shifts can drive international conflicts and are highly problematic for the communities and businesses that depend on these living marine resources. Advances in climate prediction mean that in some regions the drivers of these shifts can be forecast up to a decade ahead, although forecasts of distribution shifts on this critical time-scale, while highly sought after by stakeholders, have yet to materialise. Here, we demonstrate the application of decadal-scale climate predictions to the habitat and distribution of marine fish species. We show statistically significant forecast skill of individual years that outperform baseline forecasts 3–10 years ahead; forecasts of multi-year averages perform even better, yielding correlation coefficients in excess of 0.90 in some cases. We also demonstrate that the habitat shifts underlying conflicts over Atlantic mackerel fishing rights could have been foreseen. Our results show that climate predictions can provide information of direct relevance to stakeholders on the decadal-scale. This tool will be critical in foreseeing, adapting to and coping with the challenges of a changing future climate, particularly in the most ocean-dependent nations and communities.

[1] Danish Meteorological Institute, Copenhagen, Denmark. [2] National Institute of Aquatic Resources (DTU-Aqua), Technical University of Denmark, Kgs, Lyngby, Denmark. [3] National Center for Atmospheric Research, Boulder, CO, USA. [4] Geophysical Institute, University of Bergen and Bjerknes Centre for Climate Research, Bergen, Norway. [5] Nansen Environmental and Remote Sensing Center, Bergen, Norway. [6] Max Planck Institute for Meteorology, Hamburg, Germany. [7] Helmholtz-Zentrum Hereon, Institute of Coastal Systems - Analysis and Modeling, Geesthacht, Germany. ✉email: mapa@dmi.dk

Our current understanding of the impacts of climate change typically focuses on the climatic time scale; for example, 50 or 100 years into the future. While these timescales are of value to strategic decision-making by, for example, governments, they are far from the seasonal, annual, and decadal timescales on which regional bodies, local governments, businesses and individuals make most of their decisions[1]. The recent development of near-term climate predictions[2,3] can potentially fill this gap and examples of such climate services can already be found on the sub-seasonal and seasonal timescales[4,5]. However, on the important annual-to-decadal timescale predictive skill is currently limited to the ocean[6,7], restricting their applications. Decadal predictions of the ocean could, nevertheless, be invaluable in supporting climate adaptation and sustainable development[8] in coastal communities and nations, particularly in the Global South[9,10] where ocean-dependency and climate risks are highest[11].

Realising the potential societal benefits of oceanic decadal predictability, however, requires converting climate predictions into information that addresses the challenges directly faced by stakeholders. One such challenge is the ongoing climate-driven global redistribution of species, the largest since the Last Glacial Maximum[12]. Shifts in where species are found (i.e., their distribution) have been reported in the ocean from the lowest trophic levels to the largest top-predators[13] and are occurring faster than on land due to the higher vulnerability of marine species to warming[14]. Projections suggest that this trend will continue with impacts being felt globally[8,9,15]. As traditionally fished species disappear and new species arrive, local communities and fishers are required to adapt their fishing techniques, infrastructure, markets, traditions, and even culinary preferences to the changed fishing opportunities. International conflicts over fishing rights can also arise as shifting fish stocks start to straddle international jurisdictions, an issue that is only expected to worsen: transboundary stocks may impact as many as 40% of exclusive economic zones in the future[15]. Examples of such conflicts are already being seen[16], for example the so-called North Atlantic "mackerel war"[17] between the European Union, Norway, Iceland, and the Faroe Islands over access to Atlantic mackerel (*Scomber scombrus*), and are a leading cause of international disputes between democracies[15,16]. The ability to foresee such shifts can therefore potentially hold the key to both avoiding conflict and adapting marine fisheries to a changing climate[8,15].

Here, we describe the direct applications of decadal climate predictions to forecast shifts in the habitat and distribution of marine species. We draw a necessary distinction throughout this work between the habitat of a species (where conditions are suitable for occurrence) and its distribution (where it is actually found). We focus on three exemplar fish species in the North Atlantic that have shown well-documented distribution shifts in recent years. The Northeast Atlantic stock of mackerel supports one of the most valuable fisheries in Europe and recent distribution shifts into Icelandic and Greenlandic waters[18] have driven the aforementioned conflict over fishing rights. Atlantic bluefin tuna (*Thunnus thynnus*) is a large commercially valuable and endangered top predator: in recent years the species has shifted into the Irminger Sea and Denmark Strait[19], opening up new fishing opportunities for Iceland and Greenland[20]. Blue whiting (*Micromesistius poutassou*) has at times been one of the world's largest fisheries and its spawning distribution shifts regularly between the waters of the UK, EU, Faroe Islands and areas beyond national jurisdiction[21], a potential point of conflict in light of the UK's recent departure from the EU. For each of these cases we combined existing biological habitat models characterising the species' environment preferences[18,19,21] (Supplementary Table 1) with predictions of the physical environment from existing climate prediction systems (Supplementary Table 2) to produce decadal-scale habitat predictions. We then verified these predictions against habitat estimated from ocean observations and, to the extent possible, against observations of distribution.

## Results and discussion

We first show the ability of climate prediction systems to skilfully forecast the physical drivers that serve as inputs to the biological models on multi-annual timescales (see Methods). Habitat models show that sea surface temperature (SST) in the warmest month (August) shapes mackerel and bluefin tuna habitat while sub-surface salinity (250–600 m) during the peak spawning month (March) is the primary environmental driver shaping blue whiting spawning habitat (Supplementary Table 1). The five-year predictive skill of these variables, as assessed by performing retrospective predictions and comparing against observations, is generally high and statistically greater than zero in most parts of the domain (Fig. 1). This result is in line with more general results (e.g., annual and regional averages) reported elsewhere[7,22]. The skill also matches well with the regions relevant to each of the marine species that we consider, providing a solid base from which to develop habitat forecasts. Similar results are seen when considering the absolute error in the forecasts (Supplementary Fig. 1) rather than the correlation (Fig. 1).

The ability to forecast the state of the ocean in these regions carries over into the ability to forecast habitat on multi-annual and decadal time scales. Outputs from the climate prediction systems are used in the habitat models and their habitat predictions are compared with estimates from ocean observations. The area of suitable habitat (e.g. km$^2$) within the relevant regions of interest for each species (Fig. 1) was used as our metric of interest. Pearson correlation coefficients between the forecast habitat metrics and those derived from observations are generally high, up to 0.75 for the forecast including all ensemble members ("Grand ensemble", Fig. 2). This skill remains high even at decadal lead times, and is significantly greater than zero (p < 0.05) for all leads and species. Individual modelling systems can have lower performance, but the combination of models into a grand ensemble generally gives the best performance.

Importantly, our habitat forecast systems also outperform alternative simpler approaches. We consider first a persistence forecast, where "tomorrow is the same as today", as a much simpler and commonly used baseline system: a valuable forecast system should have skill over and above such a reference forecast[23]. For short lead times (e.g., one-two years), persistence forecasts are generally comparable to climate predictions (Fig. 2), reflecting the high inertia of the ocean. In these cases, the improvement in skill of habitat forecasts over persistence forecasts is generally not significant or at best weak. On the multi-annual timescale, however, persistence skill starts to fade while the decadal prediction systems maintain their skill. For all three fish stocks considered, forecast performance for leads of three or more years is significantly greater than persistence (p < 0.05 or better), and can therefore be considered skilful. Alternative skill metrics considering the absolute errors in the forecasts (via the Mean-Squared Error skill score, MSESS) and reliability of the predictive distributions (Continuous Ranked Probability skill score, CRPSS) also show significant skill across all fish stocks and for multi-annual to decadal lead times (Supplementary Fig. 2). As is common in multimodel ensemble systems[24], the grand-ensemble forecast based on all 85 members weighted equally is also consistently the among the best performing forecasts (Fig. 2) reiterating the value of large ensembles in climate prediction[6].

A second alternative approach to habitat forecasting could be to use climate projections directly as forecasts: such uninitialized

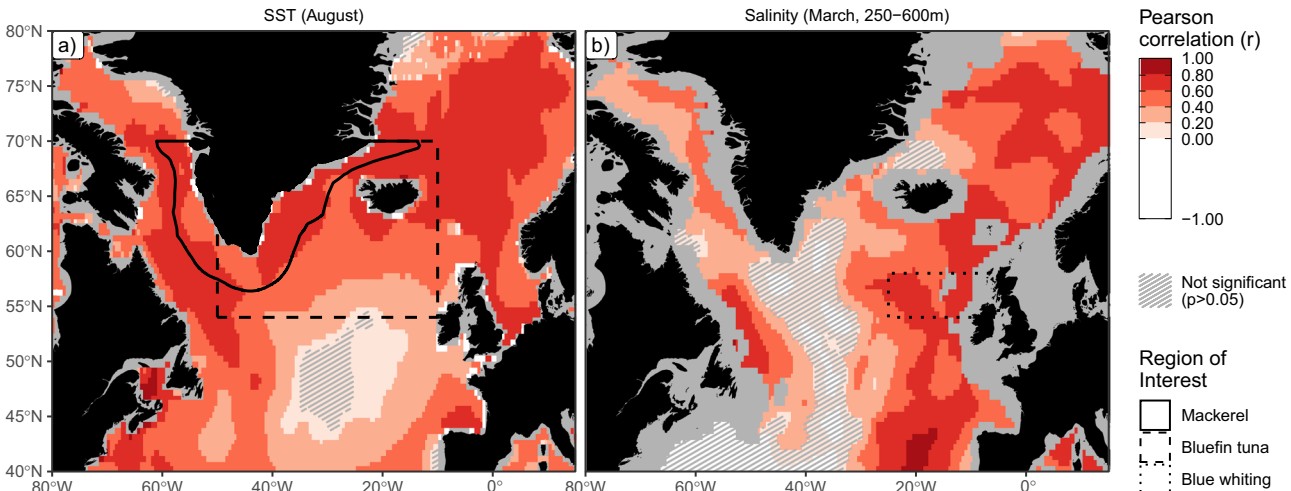

**Fig. 1 Ocean state can be skilfully forecast.** Predictive skill of physical variables underlying our habitat forecasts from climate prediction systems with a lead time of five years for (**a**) mean August sea surface temperaure (SST) and (**b**) mean March sub-surface (250–600 m) salinity. Predictive skill is expressed as the Pearson correlation coefficient (*r*) between the forecast and observed values of each variable, with each grid point coloured according to the local value, evaluated over the period 1960–2018 for SST and 1985–2018 for salinity. Forecast skill is for the grand ensemble mean forecast, i.e., a forecast averaged across the individual realisations from all model systems. Regions where the correlation coefficient is not significantly greater than 0 (at the 95% confidence level, as estimated from bootstrapping) are cross-hatched. Lines mark the polygons over which the area of suitable habitat is calculated in subsequent analyses. Ocean regions not represented by all forecast models are shown in grey.

projections are much simpler to generate and work with than the initialised climate predictions used above (where forecasts are started from estimates of the ocean state). However, similar to what is seen for forecasts of physical variables in the northern North Atlantic[3,7], our habitat forecasts based on initialised climate predictions outperform those constructed from uninitialised climate projections (Supplementary Fig. 3). The gain due to initialisation is particularly apparent when viewed from a probabilistic point of view: the ability of the forecast systems to correctly represent the habitat probability distribution (as indicated by the CRPSS metric) was significantly better than the uninitialized forecasts for all lead times and species (Supplementary Fig. 3c). This result is consistent with expectations, as the initialisation process pushes the predictions towards observations and narrows their distribution compared to uninitialized models, yielding forecasts that are both more accurate and more precise. Given the well-established need to communicate both the most likely value and the potential range of values (i.e., uncertainty) together in a forecast[25], the significantly better probabilistic performance of initialised systems makes them clearly superior to uninitialized models in these cases.

While we capture the majority of the variability when forecasting individual years, the forecasts perform even better when considering averages across multiple years. We calculated multi-year means of habitat area derived from both the climate prediction systems and observational data, and then re-evaluated the forecast skill (Fig. 3a–c). Improvements in skill are seen for all fish stocks, reaching correlation coefficients of 0.95, 0.94 and 0.74 for predictions of the decadal average (9-year window) for mackerel, bluefin tuna and blue whiting respectively. Averaging also improves some persistence forecasts but the habitat forecast system continues to be significantly better ($p < 0.01$ for 9 year averages across all stocks). The ultimate choice of averaging window clearly depends on the needs of the decision maker using the forecast: short-term tactical planning may need the individual years, while longer-term strategic planning may require the decadal averages or the statistical distribution. Importantly, and reassuringly, we show significant decadal forecast skill of both the

mean (Pearson correlation and MSESS) and the distribution (CRPSS), with and without averaging.

The effect of multi-year averaging on our predictions is closely linked to the source of their skill. On short timescales, process originating from atmospheric dynamics (e.g., blocking highs) strongly influence oceanic variability, especially for SST: while these processes are present in climate models, they are not predictable beyond weather timescales due to their chaotic nature. On longer timescales, the North Atlantic subpolar gyre, Atlantic Multidecadal Variability, and anthropogenic warming set the background oceanographic conditions on top of which high-frequency variability is imposed: the representation of all of these aspects of Atlantic thermohaline circulation benefit from the initialisation process in climate prediction models and are well predicted[3,6]. The ability to capture the lower-frequency variability of the physical system also propagates into our habitat forecasts, which are clearly better at capturing multiannual variability than interannual (e.g. Fig. 3d–f). Multiannual averaging improves these forecasts further by effectively filtering out the high-frequency interannual-noise, thereby increasing the relative contribution of the predictable low-frequency components. The skill of our habitat forecasts is therefore primarily due to the strong low-frequency (decadal) variability in the system, together with the ability of initialised climate prediction models to capture these processes, paralleling results reported elsewhere[26].

It is important to note that the biological models used here represent the habitat of each species and not their distribution. Habitat, in this context, corresponds to the spatial locations where the species could potentially be found, whereas distribution refers to where the species actually was (or will be) found. Many processes influence the way in which species do, or do not, use their available habitat, including competition, presence (or absence) of predators, schooling and migration, behavioural dynamics, and the need to reproduce[27]. Habitat is also further constrained by environmental factors that are not included in these models (e.g. food quantity and quality). Forecasts of the presence of habitat should therefore be viewed as a necessary, but not sufficient, condition to observe a species at a given location: the presence of suitable habitat does not

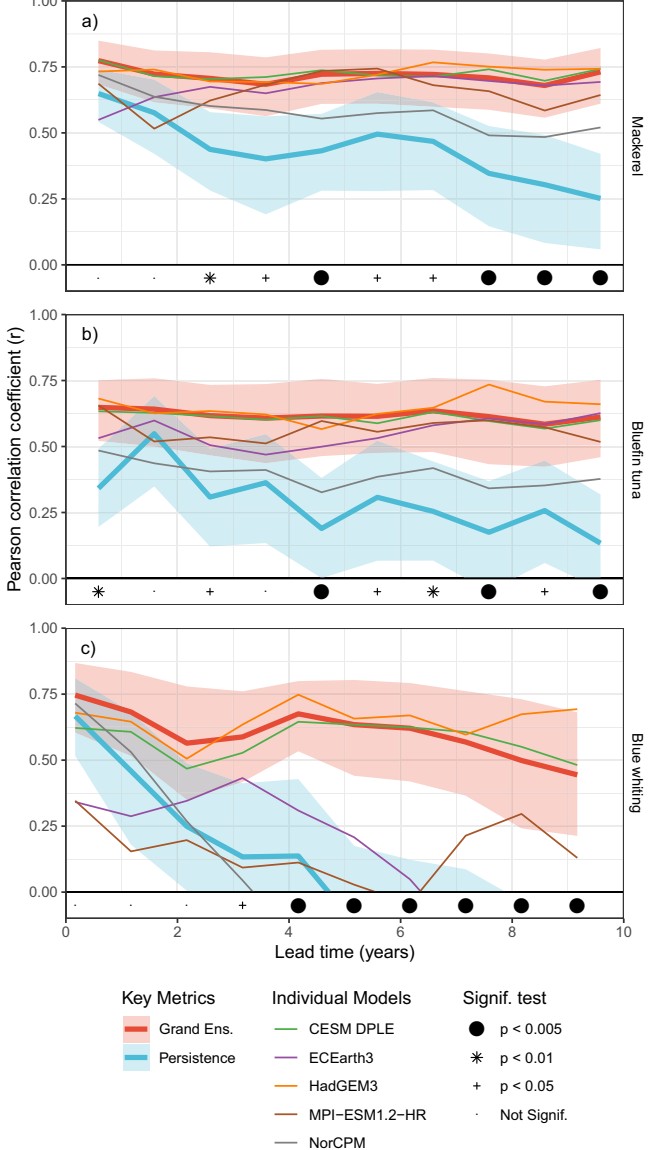

**Fig. 2 Fish habitat can be skilfully forecast.** Forecast skill for the area of suitable **a** mackerel, **b** bluefin tuna, and **c** blue whiting habitat. Forecast skill is given as the Pearson correlation coefficient (*r*) between the forecast habitat area and that derived from observational data, and is plotted as a function of forecast lead time into the future calculated across the appropriate comparison periods. Forecast skill is shown for the mean forecasts of the individual models (light coloured lines) and for the grand-ensemble forecast across all ensemble members (heavy red line). The skill of persistence forecasts (heavy blue lines) are also shown for reference. Shaded areas for both these key metrics denote the 90% confidence interval estimated by bootstrapping: 5% of the distribution is therefore above and 5% below the shaded areas. The hypothesis that the grand-ensemble forecast outperforms persistence (i.e. a one-tailed test) is tested for each lead time, and denoted with symbols at the bottom of each panel. Source data are provided as a Source Data file: exact *p* values are available in this source data.

guarantee the presence of fish. On the other hand, the absence of suitable habitat should guarantee the absence of fish. The skill of our habitat forecasts for predicting distribution shifts is therefore asymmetrical in practice because habitat models are much better at predicting absence than presence.

The recent decline in the spawning habitat of blue whiting illustrates this interplay clearly (Fig. 4b). In the mid-2000s, when the blue whiting stock was at its highest recorded level, the observed area of the distribution closely corresponded to the habitat estimated from oceanographic observations. While the amount of this habitat slowly declined (a feature predicted by decadal forecast systems), the area of the distribution collapsed much more rapidly as the stock shrank due to a high fishing pressure. Recovery of the stock was accompanied by the expansion of the distribution, but only back to 2/3 of the area seen earlier (as predicted by the forecast systems). As this example shows, habitat forecasts are best interpreted as constraints on the distribution of species.

While the ability to forecast the habitat and distribution of the fish species that they depend upon is potentially of great value to stakeholders, avoiding conflicts due to shifting distributions requires more than just reliable predictions. For example, stakeholders also need to have the ability to act on this information[1,28]. Distribution shifts will often result in both winners and losers and there is therefore a natural tendency on the part of the negatively impacted party to resist change. International agreements for managing such transboundary stocks need to have sufficient flexibility to cope with distributional shifts, while at the same time ensuring the sustainability of both the agreement and the fish stock itself[15]. Decadal forecasts of habitat and distribution can be integral to such agreements, allowing foresight and the development of adaptive measures[29].

More generally, these results also highlight the emerging potential of marine-ecological forecasting as a climate change adaptation tool[8]. While we have focused here on the North Atlantic region, annual and multi-annual forecast skill is present in many other large marine ecosystems[30] and can underpin the development of similar forecasts elsewhere[31]. This technology is also particularly relevant to Small Island Developing States (SIDS) and the Global South, where ocean dependency and climate risk are among the highest in the world[9]. Regularly produced global-scale decadal-forecasts[22] can support relevant climate services and thereby the sustainable development and climate adaptation of these nations[8], for example via the UN Decade of Ocean Science for Sustainable Development with it's clear focus on "A Predicted Ocean". Decadal-scale forecasts of the ocean, and of the life in it, thereby represent a tremendous opportunity for cutting edge climate science to have a direct benefit for the local communities, businesses and individuals that are most at risk from a changing and variable oceanic climate.

## Methods

**Study region.** We focus on the northern North Atlantic as the basis for this work. Decadal prediction experiments have shown this region to be one of the most predictable parts of the planet, especially on the decadal scale[32,33]. Studies have shown multi-annual to decadal predictability for sea surface temperature[34], upper ocean heat content[35], the Atlantic Meridional Overturning Circulation (AMOC)[36], $CO_2$ uptake[37], and the dynamics of the North Atlantic subpolar gyre[7,38,39]. This high underlying predictability of the physical system makes the North Atlantic an ideal candidate in which to develop decadal ecological forecasts and climate services[30,31].

**Fish species and habitat models.** We focus on three case studies in the North Atlantic region (Supplementary Table 1); mackerel (*Scomber scombrus*), bluefin tuna (*Thunnus thynnus*) and blue whiting (*Micromesistius poutassou*). The choice of these fish species was guided by several factors that we expect will make their habitats amenable to prediction[31]. Firstly, as noted above, the species studied are present in one of the most predictable regions of the global ocean. All of these species have also shown significant, well-documented shifts in their spatial distribution in recent decades and there is furthermore an established body of knowledge characterising the mechanisms and drivers underlying each of these shifts. Most importantly, habitat models (also known as ecological niche or species distribution models) parameterising the relationship between observations of these

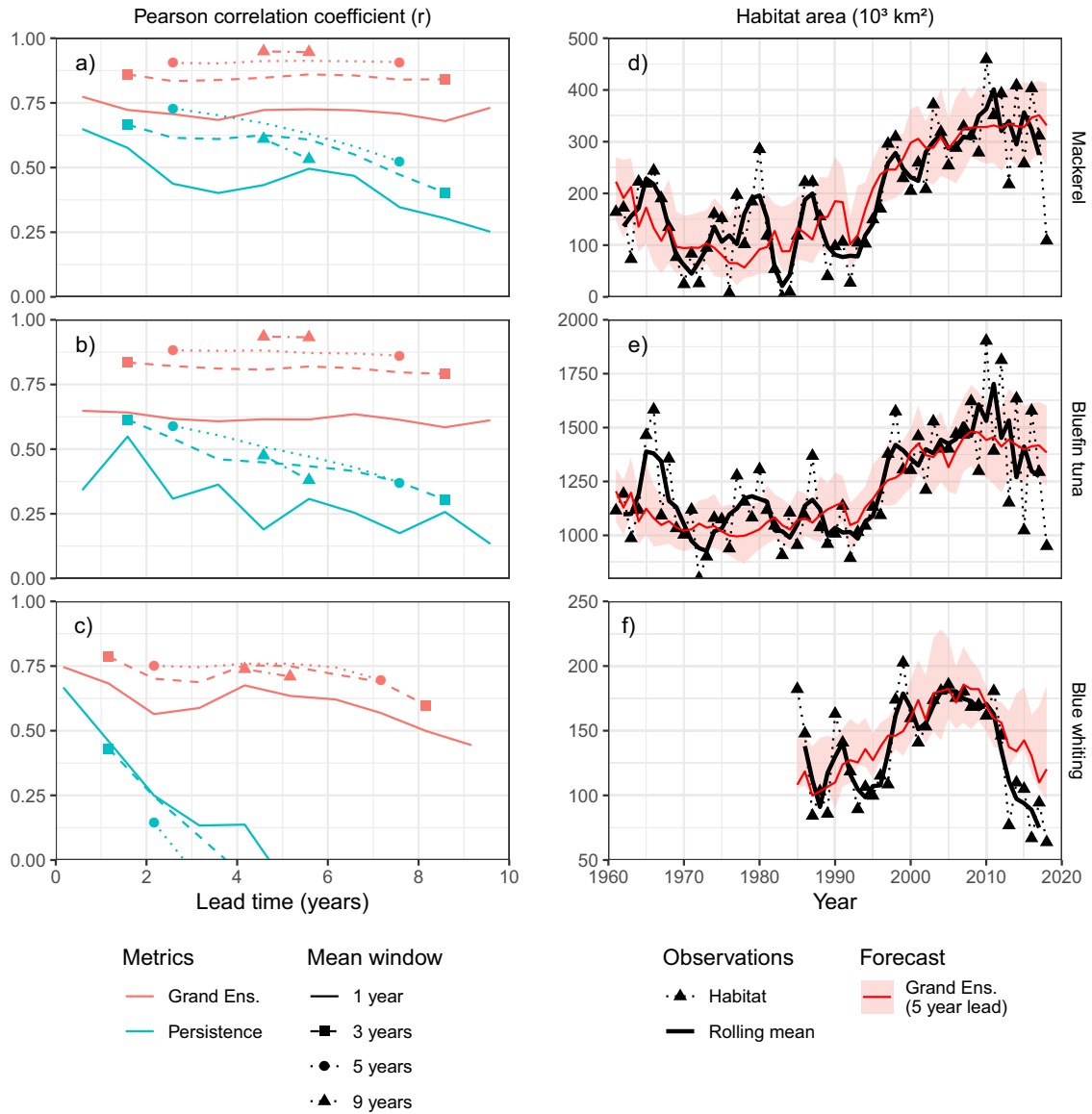

**Fig. 3 Habitat forecasts capture low-frequency variability.** The forecast skill of multi-annual averages of habitat area (panels **a-c**), as characterised by the Pearson correlation coefficient (*r*), is shown for the grand-ensemble and persistence forecasts. In addition to the single-year values also plotted in Fig. 2 (solid lines), the skill of multi-year averages (3, 5, and 9 year centred means) are also shown (broken lines with symbols). Lead-time is defined as the length of time from the issuing of the forecast (1 January) to the middle of the running mean window. Multiyear forecasts are significantly better than multiyear persistence for all lead times (*p* < 0.01, one-tailed test, as estimated by bootstrapping). Time-series of habitat metrics (panels **d-f**) show habitat estimates based on observations (triangles connected by dotted line) with their three-year running means (solid black lines). Habitat metrics forecast by the grand-ensemble (solid red line) with a 5-year lead time are shown with the corresponding 90% range of realizations (shaded area). Time series are shown for the full range of years used to estimate the forecast performance (i.e., 1961–2018 for mackerel and bluefin tuna, 1985–2018 for blue whiting). Panels (**a**) and (**d**) show results for the area of mackerel habitat around south Greenland, panels (**b**) and (**e**) bluefin tuna habitat south of Iceland, and (**c**) and (**f**) blue whiting spawning habitat west of Great Britain and Ireland. Source data are provided as a Source Data file.

species and the physical environment already exist. We discuss each of these case studies in turn below.

Recent shifts in the distribution of mackerel are amongst the most well-known examples of fish distributional shifts. The feeding distribution of mackerel expanded northwards and westwards to Iceland in 2007[40] and Greenland in 2011[18], leading to international conflicts over fishing rights on this species[17]. A wide variety of explanations for these shifts can be found, with the effects of climate change and density-dependent expansion being the most common[41–44]. However, the distribution of mackerel is clearly limited by temperature, with 8.5 °C serving as a lower threshold[18,41]. We therefore used the 8.5 °C August-mean isotherm as a threshold for the habitat of this species. Paralleling other studies with a clearly defined habitat model[18], we focused on the waters around Greenland (specifically the exclusive economic zone south of 70 °N): being at the cold thermal range limit of mackerel, the temperature is expected to have a controlling influence on habitat variations in this region.

Bluefin tuna are pelagic top predators that are widely distributed throughout the North Atlantic. The thermally suitable feeding habitat of this species expanded by 800 000 km² from the mid-1980s to the early 2010s, leading to the first documented observation of the species in Denmark Strait in 2012[19]. Unusually for fish, Bluefin tuna have the ability to regulate their body temperature: their core temperature is therefore often above the surrounding waters. Data-storage tags measuring both internal and external temperatures show that the species can dive into colder waters during the day for short periods to feed (e.g., horizontally across fronts or vertically across the thermocline), during which time the core temperature starts to drop, but then return to surface waters during the night to warm-up again[45]. Such studies suggest that the species therefore needs access to surface waters of at least 10–11 °C to support foraging, which can be interpreted as a natural limitation on the distribution of the species and definition of habitat[19]. These conclusions are also seen in the results of empirical habitat models[46–48] that arrive at similar thresholds based on observations. Entirely independently,

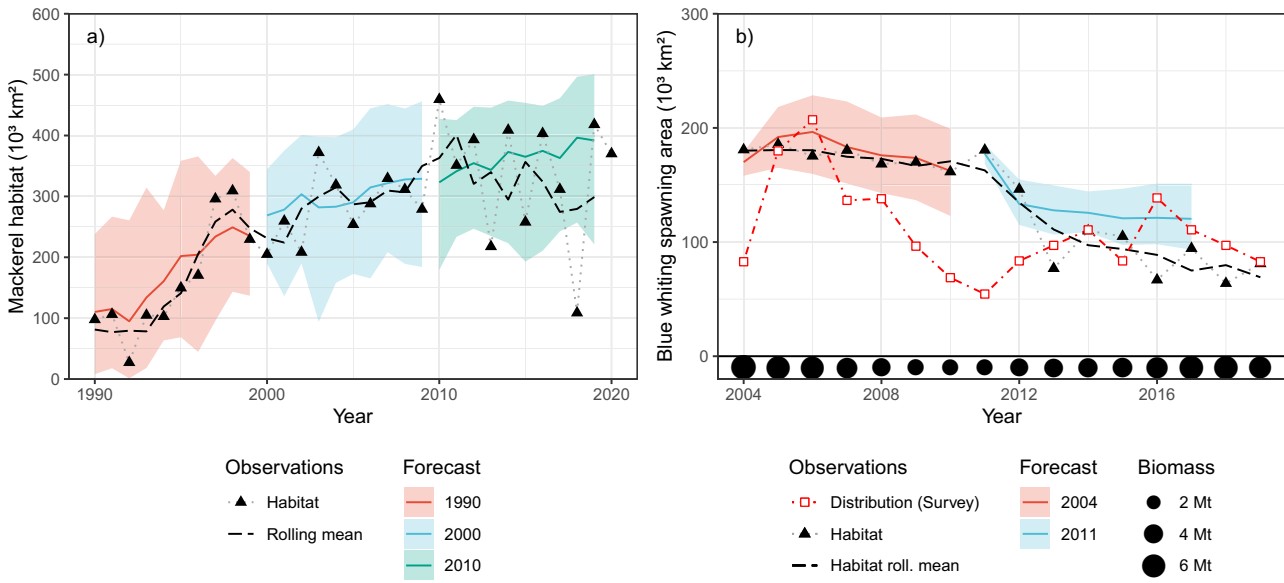

**Fig. 4 Key habitat and distribution shift events can be forecast.** Habitat forecasts from the best performing individual model system (CESM-DPLE) started at select times (solid coloured lines) are shown with the corresponding 90% range of realizations in this model (shaded colours area) for the area of (**a**) mackerel habitat around south Greenland and (**b**) blue whiting spawning habitat. Habitat metrics based on observed conditions (see Methods) in a given year are shown (triangles connected by dotted grey line) together with a three-year centred running mean (dashed black line) of these values. For (**b**), blue whiting, the distributional area estimated from scientific monitoring surveys (red dashed line) is shown together with the stock biomass (bullets at bottom of figure) estimated by the stock assessment. An illustrative subset of predictions is shown on both panels: the full set for a 5 year lead-time can be seen in Fig. 3d–f. Source data are provided as a Source Data file.

mechanistic bioenergetics modelling of the oxygen requirements and aerobic capacity of the species also reached a similar result[46]. Like others[19,20], we therefore employ the 11 °C isotherm for the August mean (the warmest month in the region) to define the cold-limit of thermally-suitable feeding habitat for this species in the northern North Atlantic.

Blue whiting is a small mesopelagic species found widely throughout the eastern North Atlantic. The species supports a large commercial fishery, primarily for industrial uses, that has varied greatly over time: in 2004 it was the world's third-largest fishery, with catches of 2.4 million tonnes[49]. While smaller sub-populations exist, the largest stock, and the one that supports the majority of the fishery, migrates between its feeding grounds in the Norwegian Sea and spawning grounds to the west of Great Britain and Ireland in the Rockall Trough region. Spawning takes place from February to April[50] and the spawning distribution varies substantially between years, expanding and contracting on and off the Rockall Plateau[51]. Initial work linked these changes to the large-scale dynamics of the North Atlantic sub-polar gyre[52]; however, more recent work has narrowed this view down to the local salinity conditions[21] (which in turn are shaped by the basin-scale dynamics of the gyre). This work was based on approximately 34 000 observations (1100 presences) of blue whiting larvae in this region from 1951 to 2005 from the Continuous Plankton Recorder (CPR), which, in addition to the planktonic species for which it is best known, also regularly captures fish larvae[53,54]. A habitat model has been developed and parameterized based on this data, using latitude, day of year, bathymetry, the solar elevation angle and environmental variables (averaged over 250–600 m) as predictors. The likelihood of observing blue whiting larvae in the CPR was found to have a dome-shaped response to salinity, with larvae occurrence limited to salinities between 35.3 and 35.5 psu. This model shows good agreement with independent observations from both scientific surveys and the fishery on the stock, and currently forms the basis for operational forecasts of the spawning distribution[55]. The full habitat model[21] was applied here to define suitable spawning habitat for this species, but was focused on the northern component of the stock[50] where most of the variability has been observed.

**Physical observations.** Two different datasets were used as the basis for characterizing the physical environment. Sea surface temperature estimates were based on the HadISST v1.1 product[56], while sub-surface salinity estimates were based on the EN4 product (v4.2.1 analysis, with Gouretski and Reseghetti's corrections to the source profile data[57]), both from the UK Met Office. Both products are high-quality, internationally recognized estimates of the state of the ocean covering an extended time period (HadISST: from 1860, EN4 from 1900) and are presented on a regular 1° grid as monthly averages.

**Decadal forecast models.** An ensemble of five decadal prediction systems was collated for this analysis: all models followed the CMIP6 Decadal Climate-

Prediction Project (DCPP) protocol[58]. For each decadal prediction system, a database of retrospective forecasts was available based on annual initialisations. For each of these initialisations, a fully-coupled (ocean, sea ice, land, atmosphere) model was run using the CMIP6 historical (for years 1960–2014) and ssp2-4.5 (for years 2015–2030) forcing from the given starting point to generate forecasts up to 10 years after the initialisation. Multiple realisations were available for each of the initialisations for each of the model systems, to give a grand ensemble of 85 members. Details are given in the relevant references (Supplementary Table 2) and the DCPP protocol[58]: for a more general introduction to climate prediction and decadal forecasts, several references are recommended[3,33,59].

**Uninitialized projections.** We used uninitialized historical and projected climate simulations from CMIP6[60] as an additional form of reference forecast. We selected a temporal subset of SST and salinity model outputs for the "historical" (covering the years 1960–2014) and "SSP5-8.5" (2015–2020) experiments. We used one realization from each model system, with the "Variant Label" identifier being maintained between the two experiments. Only models that fully covered the comparison period (1960–2018) were retained. Model outputs presented on sigma or density vertical axes or unstructured horizontal grids were excluded due to difficulties in incorporating them into the processing chain. Native model resolution (grid label "gn") was used as the first preference, where available, followed by lower-resolution regridded products ("gr","gr1"). After this selection process, 35 models of salinity and 44 models of SST were incorporated into the analysis (Supplementary Table 3). Details of these experiments are given in the CMIP6 protocol[60].

**Data processing.** Model and observational ocean data (temperature and salinity) were processed in the same manner. Data stored at multiple model levels (i.e., subsurface salinity) were first extracted and then averaged (weighted by layer thickness) over the 250–600-m depth range to produce two-dimensional fields for each time step on the native model grid. The months of interest were then extracted from all fields and regridded using bilinear interpolation onto a common regular 0.5° latitude-longitude grid covering the regions of interest (Fig. 1).

**Observational climatologies.** Extracted and regridded observational data were used to generate monthly climatologies by grid-point averaging based on the 30-year period from 1985 to 2014 (inclusive). This period was chosen to cover the first 30 years for which predictions were used for all three fish case studies.

**Bias correction.** Processed model outputs were bias-corrected. Climate model outputs often show spatially variable systematic biases relative to observed values. In the case of climate prediction systems, these biases can also vary as a function of forecast lead time due to the forecast drift phenomenon associated with adjustment

from the initialized state to the model climate[61]. All decadal forecast models used were initialised once per year on the same start date, so it was not necessary to include start-date dependent bias-corrections (as would be appropriate for forecasts started e.g. every month). Model outputs were bias-corrected following the full field approach, irrespective of the model initialisation technique applied[62]. Briefly, the climatological field of each variable (salinity and SST) was calculated for a given model and forecast lead time by grid-point averaging over the same 30-year period as the observational climatology. The individual members of the forecast ensemble were then converted into a forecast anomaly based on this climatology for a given lead-time. Bias-adjusted full-field forecasts were then produced by adding the appropriate observational climatology to the forecast anomaly.

**Ensemble means**. Mean forecasts of environmental parameters for each model system were produced by averaging the forecast fields across the realisations for that forecast model. The mean across all 85 realisations in the ensemble was also calculated to produce a grand-ensemble mean forecast ("Grand Ens."): in this way, each realisation was given equal weighting in the forecast, irrespective of the model system it came from or the number of siblings it may have. We also considered the mean-of-model-means approach, where the mean-forecasts from each climate prediction system were averaged, thereby giving each model system equal weight. However, the performance of this approach was indistinguishable from (or worse than) the simpler grand ensemble approach, and is not presented here.

**Habitat metrics**. The skill of the near-term forecasts was evaluated based on retrospective forecasting (also known as hindcasting, although the meaning of this term can differ between fields and so is avoided here). The habitat models described above were applied to the bias-corrected full-field forecasts to generate a comparable set of retrospective habitat forecasts. Where the habitat model gave a binary outcome (suitable / unsuitable habitat), the total area of suitable habitat was calculated directly. The blue whiting habitat model[21] however returns the probability of observing larvae and calibration is therefore needed prior to calculating the spawning habitat area for the observation of adult fish[63]: the threshold was chosen to ensure agreement between the upper quartile of the annual adult distribution areas observed and the corresponding habitat estimates. The habitat models were also applied to observational datasets to generate observationally-based estimates of the area of habitat in a given year ("observed habitat" in Figures).

**Persistence and binned forecasts**. Persistence forecasts are used as the primary choice of reference forecast: a forecast is viewed as skillful if it can outperform such a baseline[23]. Persistence forecasts were generated by propagating the habitat metrics calculated in a given year forward for up to the maximum 10-year forecast horizon considered here. Binned forecasts (i.e., the average over a multi-year period) were calculated based on 3, 5, and 9-year windows of habitat metrics from all data sources (observations, persistence, uninitialized models and forecast models), and assigned a forecast time corresponding to the centre of the averaging-window. The skill of both binned and persistence forecasts was then assessed in the same manner as for other data sources.

**Forecast verification and skill**. Forecast skill was assessed by comparing the estimates of habitat based on observed environmental variables with forecasts of habitat based on the various approaches. The retrospective forecast databases available differ in their length, and a common comparison period was chosen as 1961–2018 (inclusive) for SST-based variables, based on available coverage across all models. Initial explorations of salinity forecasts, however, revealed substantial inconsistencies in the region of interest between observational products prior to the mid-1980s that propagated into the initial conditions used to initialise the climate predictions. In the absence of agreement between observational products about salinity in this region, we therefore limited the comparison period for salinity (and salinity-based habitat models) to 1985–2018 (inclusive).

Forecast skill was quantified using multiple metrics[23,64] including the Pearson correlation coefficient ($r$), the Mean Squared Error skill score (MSESS) and the Continuous Ranked Probability skill score (CRPSS) between observed and predicted habitat metrics[23]. Skill-scores were calculated relative to the mean and standard deviation of the habitat metrics over the climatological period. CRPSS scores were calculated for the grand ensemble by considering the forecasts across all 85 ensemble members. Confidence intervals around each of these metrics were generated for each lead time by a bootstrapping approach i.e. 1) pairwise resampling of years with replacement; 2) recalculating the appropriate metric; and 3) repeating the process 1000 times. We then interpret these confidence intervals in terms of a one-tailed test, i.e. is the metric significantly greater than zero. In cases where two skill metrics are compared (e.g. persistence correlation with forecast correlation), we use the 1000 samples of each skill metric in a pairwise manner, comparing the individual values to calculate the proportion of times that one metric exceeds the other, which we interpret as the significance level.

**Distribution and abundance data**. Observations of distribution shifts suitable for verifying forecasts can be challenging to obtain, particularly for widely distributed species such as those considered here. While we were unable to find suitable

scientific monitoring datasets for bluefin tuna and mackerel, the sudden appearance of these species beyond their traditional range has been well documented in the scientific literature (see "Fish Species and Habitat Models" section above).

The distribution of blue whiting, however, has been the subject of routine scientific monitoring surveys since the early 1980s: since 2004 these surveys have been coordinated and standardised as the International Blue Whiting Spawning Stock Survey[65]. Observations of blue whiting from this survey on a regular 2° × 1° grid were first used to identify the core 99% of the distribution in each year and the area occupied was then calculated. Estimates of the spawning stock biomass were obtained from the ICES Standard Graph Database (http://standardgraphs.ices.dk/) for the most recent stock assessment performed in 2020 (ICES SAG assessment key 13880).

**Reporting summary**. Further information on research design is available in the Nature Research Reporting Summary linked to this article.

## Data availability
Climate predictions and projections analysed in this study are available from the CMIP data archives https://esgf-node.llnl.gov/projects/cmip6/. CESM-DPLE data are available from http://www.cesm.ucar.edu/projects/community-projects/DPLE/. HadISST data is available from https://www.metoffice.gov.uk/hadobs/hadisst/ and EN4 data from https://www.metoffice.gov.uk/hadobs/en4/. ICES stock assessment data is available from http://standardgraphs.ices.dk/. Figure source data are provided with this paper.

## Code availability
The code used during the current study is available online: https://github.com/markpayneatwork/PredictabilityEngine (https://doi.org/10.5281/zenodo.6451271)[66].

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

## Acknowledgements

The CESM project is supported primarily by the US National Science Foundation (NSF). The National Center for Atmospheric Research (NCAR) is a major facility sponsored by the US NSF under Cooperative Agreement 1852977 and the US NSF Collaborative Research Grant OPP-1737377. Computing and data storage resources for CESM-DPLE simulations, including the Cheyenne supercomputer (doi:10.5065/D6RX99HX), were provided by the Computational and Information Systems Laboratory (CISL) at NCAR. We acknowledge the World Climate Research Programme, which, through its Working Group on Coupled Modelling, coordinated and promoted CMIP6. We thank the climate modeling groups for producing and making available their model output, the Earth System Grid Federation (ESGF) for archiving the data and providing access, and the multiple funding agencies who support CMIP6 and ESGF. The research leading to these results has received funding from the European Community's Seventh Framework Programme (FP7 2007–2013) under grant agreement No 308299 (NACLIM) (MRP, AKM) and the European Union's Horizon 2020 research and innovation programme

under grant agreement No 727852 (Blue-Action) (all authors). Part of the funding for this work is provided by the Danish State through the National Centre for Climate Research (NCKF) (MRP, SY). The research leading to these results has also received funding from the German Federal Ministry of Education and Research (BMBF) through the JPI Climate/JPI Oceans NextG-Climate Science-ROADMAP (FKZ: 01LP2002A) (DM) and Norwegian Grant 316618/JPIC/JPIO-04 (NK). NK was supported by the Trond Mohn Foundation (grant BFS2018TMT01; Bjerknes Climate Prediction Unit) by RCN Nansen Legacy Project (grant 276730) and by UNINETT Sigma2 AS–the Norwegian national computing facility (NN9039K, NS9039K).

## Author contributions

M.R.P. conceived the idea, performed the analyses and drafted the manuscript. A.K.M. and M.R.P. developed the blue whiting habitat model. N.K., S.G.Y., S.Y., G.D. and D.M. provided climate prediction model outputs and advised on their integration into the analysis. All authors interpreted and discussed the results, contributed to revision of the manuscript and have approved the final version. Author-order after the first author is alphabetical by last name.

## Competing interests

The authors declare no competing interests.
