## [Peer Review File · Nature Communications]

REVIEWER COMMENTS

Reviewer #1 (Remarks to the Author):

In this paper, the authors apply decadal-scale climate predictions to forecast habitat shifts of marine fish, using the examples of the northeast Atlantic mackerel, the Atlantic bluefin tuna and the Blue Whiting. Overall, I find this paper novel, insightful and very valuable. The topic this paper addresses is extremely relevant and timely. The paper is structured appropriately and overall well-written.

I have just two general suggestions for the authors to consider. I then present a few more specific comments indicated by the line number. Overall, I suggest publication after addressing those comments deemed relevant below.

General suggestions

First, the authors indicate starting from line 216 that they use models to predict changes in habitat, not distribution, and clearly explain the important difference between the two concepts. I suggest that this caveat is moved to the front of the paper, and that the remaining text is adjusted to reflect that the authors are not attempting to model distribution, but habitat. Now there are several lines in the paper that would suggest otherwise (including the title), and I suggest adjusting the language throughout. Some examples: lines 34, 76 (remove distribution), line 107 (are these not primarily 'habitat drivers' then as there are many more distribution drivers not included here as explained by the authors?), lines 227, 252 etc.

However, if there is some overlap between habitat and distribution modelling (which there seems to be), I suggest explaining more clearly where the models used by the authors situate themselves in that space.

Second, I would suggest a bit more clarity on why the following elements were selected:

- a) Why this region?
- b) Why these species?
- c) Why salinity and SST for blue whiting and mackerel/bluefin respectively?
- d) Why the particular months (i.e. August and March)?
- e) Why this particular region for the mackerel?

I can see that much, if not all, of this information is located in the supplementary material, but just a brief justification would be useful for the reader. For a), it seems that this region was most predictable and therefore a most useful case for your modelling purposes (from lines 368-369), but this would potentially be more difficult in other regions? I suggest briefly discussing this (i.e., the applicability of this method in data-poor regions or less predictable areas). For b), these species were likely chosen primarily for their (large) shifts in distribution? I suggest clarifying that. For c) and d), I can see that this is linked to a literature review on the most important habitat drivers for the species. I would suggest just stating this briefly (e.g., in line 98: "from our review/ from recent findings, this is the primary environmental driver..."), so the non-expert reader knows why you have made this choice. Lastly, for e), I am wondering why the authors decided to focus on the area around Greenland (from Fig1), not Iceland, as it is the Icelandic participation in the mackerel fishery (in its own EEZ) that has sparked the

conflict with the EU, Norway and Faroe Islands.

Specific comments

I.92: Is it possible for the authors to indicate for how many years/species they have observations of distribution?

I.212: I am wondering about the reliability of scientific monitoring surveys to estimate distributional area, as often areas where species could be are not monitored if it is a new distribution pattern. Could this throw off some of the models' predictive capacity?

I.240: Could the authors explain 'forecast initializations' briefly for the non-experts?

Reviewer #2 (Remarks to the Author):

I was extremely excited about the invitation to review this manuscript. The field of ecological forecasting is in its infancy, particularly in the marine domain, and this manuscript is the first (that I know of) to show skillful decadal-scale predictions of ecological metrics. This multi-annual time horizon is a critical missing gap in ocean predictability yet it is the time horizon most often sought after by decision makers. This manuscript is a huge scientific advance, shows novel results, and I anticipate will be a pioneering manuscript for the field of ecological forecasting. I congratulate the authors on a well-written and exciting manuscript.

In summary, the manuscript shows decadal predictability of habitat for three marine species of fisheries interest, mackerel, bluefin tuna, and blue whiting. The authors use output from the CMIP6 decadal prediction system as the forcing for their retrospective forecasts, and discuss the implications of an operational product for fisheries stakeholders. Overall, I thought the manuscript was well written, the methods sound, and the results offered a huge advancement in our understanding of ecological predictability. I have some minor points below that I think, if addressed, could be helpful to improve clarity for a future reader.

First, I didn't fully appreciate the value of the prediction (initialized forecasts) versus projection (uninitialized forecasts) comparison. The primary text (ln 239-250) is fine but it doesn't really consider that predictions and projections offer vastly different outlooks by design, the latter being a scenario planning exercise. I can't imagine anyone using a projection as a prediction but perhaps I misunderstand. I feel like this part of the paper (i.e. ln 239-250 + extended figure) detracts from the main message of skillful decadal predictions. If you need additional space (pending reviews) then I would opt to remove this section, but I appreciate you may disagree with my perspective. If it stays in then I suggest adding a sentence or two better describing the need for this comparison.

Second, it wasn't until I reached the methods section that it became apparent that the 'habitat models' for bluefin and mackerel were not an ecological model, but rather the area of a thermal preference. I don't have an issue with the metric itself, but I think it could be made clearer in the main text (perhaps

ln 95-97) that you are using an isotherm threshold to identify species habitat, which is distinct from the blue whiting approach that uses a habitat model. To me this is an important distinction because physical-only forecasts are inherently more skillful as they don't integrate the uncertainty associated with an ecological model.

Third, I really valued the integration and comparison with a persistence forecast. While I think comparing with persistence is important, I'm under the impression it's also general practice to detrend results to separate skill from the climatology. I.e. how well do the models perform at predicting anomalies rather than actual SST. It seems like MSESS might be doing this (I'm not familiar with this metric) so perhaps no additional work is needed, but it might be useful to add a sentence somewhere indicating that you have skill above the climatology and above persistence (say somewhere in the paragraph from ln 126-141).

Finally, I had some trouble understanding the Pearson correlation significance threshold that you used (e.g. Figure 1). Typically, I've seen significance reported from the Pearson correlations itself, not whether the correlation is great than 0. For example, for 59 years of SST an r of ≥ 0.26 is significant; for 34 years of salinity a r of ≥ 0.34 is significant. Could you perhaps explain your approach more fully in the methods somewhere?

Line-specific comments:

- Ln 30-35: This is fine but I actually think it undersells the advancement you offer to ecological forecasting. You don't have to make any changes, but I do think you could take the opportunity to more strongly spell out that (1) decadal ecological forecasts haven't been shown before; (2) that multi-annual forecasting is highly desired by end-users, and more so than at any other lead time; (3) that there is a novel research gap that needs to be filled for this specific forecast horizon.
- Ln 76-78: I thought this sentence could be made a bit more punchy and less verbose. For example: "Here we demonstrate one of the first examples of a decadal forecast of shifts in the habitat and distributions of marine species"
- Ln 98-99: 'five-year ahead predictive skill' is a bit awkward. Why not say 5-year forecast skill? Or use 'lead time' terminology?
- Ln 118: add units here for the area of suitable habitat
- 165-179: great paragraph! I really like how you identified the mechanism of predictability here. If useful, Brodie et al. 2021 also showed that species distributions are more predictable at these low frequency scales as well.
- Methods: Great – well detailed and laid out nicely!

Brodie, S., Abrahms, B., Bograd, S. J., Carroll, G., Hazen, E. L., Muhling, B. A., ... & Jacox, M. G. (2021). Exploring timescales of predictability in species distributions. *Ecography*.
<https://doi.org/10.1111/ecog.05504>

I look forward to seeing this published.

Stephanie Brodie

Reviewer #3 (Remarks to the Author):

This paper describes an assessment of skill for multiyear predictions of monthly averaged ocean properties within domains relevant to fish habitat, and hence fisheries, of the North Atlantic. The authors use a "grand ensemble" of model results for their skill assessment, and the size of that ensemble is large enough to be statistically meaningful. They include a comparison of model-based forecasts with persistence forecasts, to demonstrate the added value of the model results. Their focus on the North Atlantic is sensible, as decadal forecasting results are especially skillful in that region (such skill varies widely over the world ocean). I appreciated the discussion of potential advantages/pitfalls of using these forecasts in real-world fisheries management. I also liked the declarative style of the captions.

I was very much impressed by the writing and presentation of this work. I have no objection to publication of the manuscript in its present form. I think this is a valuable contribution which will encourage stakeholders to utilize these emerging predictive tools, as they demonstrate some genuine skill. I do have a few comments and suggestions, as follows:

Fig. 1. Can the authors shed any light on why the pattern of skill looks as it does? In particular, is there some correlation between skill and bathymetric depth? They do note some of the general processes leading to skill in the paragraph beginning at line 168, but the reader may wonder what factors lead to enhanced/degraded skill on some areas, as opposed to it just being a random artifact of a necessarily limited (although reasonably large) ensemble size.

I. 137 "skilful" should be "skillful"

I. 203 I agree that such forecasts would have been useful had they existed in 1990, but of course the technology (modeling capability) was not available at that time. But no objection to using this as an example.

I. 243 and I. 465 Predictions vs projections - It seems obvious to this reader that a forecast initialized with observed/reanalyzed ocean conditions will do vastly better than one initialized with an alternate realization of the "present" ocean, and that one would expect that advantage to decrease over time as the initial conditions are forgotten by the system. Apparently this type of comparison ("initialized" vs. "uninitialized") is standard in the field (e.g. as described in the Yeager et al. 2019), and it is potentially useful to see it quantified. I would prefer a bit more detail about what forcing applies to both runs (presumably, greenhouse gases and variations in the solar constant), what biases might be shared by both runs (e.g. climatological drift), and how the "uninitialized" runs themselves were spun up. As it stands, I needed to consult with Yeager et al. reference for these important details.

I. 489 (Bias correction section) - I assume the climatological field of each variable calculated as a function of model and forecast lead time *and* start date (day of year) of the forecast. If so, please state

explicitly.

I.512 "hindcasting" - many oceanographers use this term to mean atmospherically-forced simulations of past ocean states, so I am glad to see the authors avoid using that term for the retrospective forecasts.

Reply to Editor and Reviewers

In the following point-by-point response we have included the editors / reviews comments in blue italics, and our reply in black, normal type.

Dear Dr Payne,

Thank you again for submitting your manuscript "Skilful decadal-scale prediction of fish habitat and distribution shifts" to Nature Communications. We have now received reports from 3 reviewers and, on the basis of their comments, we have decided to invite a revision of your work for further consideration in our journal. Your revision should address all the points raised by our reviewers (see their reports below).

When resubmitting, you must provide a point-by-point response to the reviewers' comments. Please show all changes in the manuscript text file with track changes or colour highlighting. If you are unable to address specific reviewer requests or find any points invalid, please explain why in the point-by-point response.

Important: In addition to the above, you must comply with the following editorial requests; we will not be able to proceed with your revised manuscript otherwise. Please also see the Nature Communications formatting instructions, which you may find useful while preparing your revised manuscript.

POLICIES AND FORMS REQUIRED FOR RESUBMISSION

** Please complete or update the following checklist(s) to verify compliance with our research ethics and data reporting standards. Address all points on the checklist, revising your manuscript in response to the points if needed.*

The form(s) must be downloaded and completed in Adobe Reader rather than opened in a web browser. Each form must be uploaded as a Related Manuscript file at the time of resubmission.

Editorial policy checklist:

<https://www.nature.com/documents/nr-editorial-policy-checklist.pdf>

Reporting summary:

** If your paper uses custom code/software, please complete the following code and software submission checklist and make your code available for reviewer assessment, if you have not already done so. The code/software can be provided in a zip file with a readme.txt file or other instructions for installing and running the software. If appropriate, also provide example data and expected output. If you have any issues with the file upload, please let me know.*

<https://www.nature.com/documents/nr-software-policy.pdf>

We have updated all (three) forms as part of this revision.

DATA AND CODE AVAILABILITY

** All Nature Communications manuscripts must include a "Data Availability" section after the Methods section but before the References. If any of the data can only be shared on request or are subject to restrictions, please specify the reasons and explain how, when, and by whom the data can be accessed. For more information on this policy and a list of examples, see:*

<https://www.nature.com/documents/nr-data-availability-statements-data-citations.pdf>

The data availability section was already included in the first submission of the manuscript and has been updated in this revision.

** Please also include a “Code Availability” section after the “Data Availability” section. If the code can only be shared on request, please specify the reasons. For more information on our code sharing policy and requirements, please see:*

We have updated the Code Availability section to link to our GitHub repository. This repository is now publically visible.

** We strongly encourage you to deposit all new data associated with the paper in a persistent repository where they can be freely and enduringly accessed. We recommend submitting the data to discipline-specific and community-recognized repositories; a list of repositories is provided here:*

<http://www.nature.com/sdata/policies/repositories>

There is no new data generated in this manuscript.

** To maximise the reproducibility of research data, we strongly encourage you to provide a file containing the raw data underlying the following types of display items:*

- Any reported means/averages in box plots, bar charts, and tables*
- Dot plots/scatter plots, especially when there are overlapping points*
- Line graphs*

The data should be provided in a single Excel file with data for each figure/table in a separate sheet, or in multiple labelled files within a zipped folder. Name this file or folder ‘Source Data’, and include a brief description in your cover letter. The “Data Availability” section should also include the statement “Source data are provided with this paper.”

To learn more about our motivation behind this policy, please see:

<https://www.nature.com/articles/s41467-018-06012-8>

We have generated the “Source Data” file for the line graphs shown in the manuscript, and included it with the resubmission. We have also added the suggested text to the Data Availability section.

ORCID

** Nature Communications is committed to improving transparency in authorship. As part of our efforts in this direction, we are now requesting that all authors identified as ‘corresponding author’ create and link their Open Researcher and Contributor Identifier (ORCID) with their account on the Manuscript Tracking System prior to acceptance. ORCID helps the scientific community achieve unambiguous attribution of all scholarly contributions.*

You can create and link your ORCID from the home page of the Manuscript Tracking System by clicking on ‘Modify my Springer Nature account’ and following these instructions. Please also inform all co-authors that they can add their ORCIDs to their accounts and that they must do so prior to acceptance.

If you experience problems in linking your ORCID, please contact the Platform Support Helpdesk.

The corresponding author has associated their ORCID with this submission and multiple authors have also connected their own ORCID numbers.

HOW TO SUBMIT

Please use the link below to submit the following items as separate documents:

- Revised manuscript
- Any supplementary files
- Point-by-point response to the reviewers' comments, reproduced verbatim
- Cover letter to the editor
- Any completed checklist(s)

We would normally ask to see a revised version of this paper within three months but we appreciate revisions may take longer than usual and can extend this timeline if the Covid-19 pandemic prevents you from undertaking any further work for a longer period - please do get back to us on this nearer the time.

When evaluating your revised manuscript, we will not consider any similar papers published in the meantime to compromise the novelty of your study. See here for more information.

Best regards,

Iain

Iain Dickson
Editor
Nature Communications

REVIEWER COMMENTS

Reviewer #1 (Remarks to the Author):

In this paper, the authors apply decadal-scale climate predictions to forecast habitat shifts of marine fish, using the examples of the northeast Atlantic mackerel, the Atlantic bluefin tuna and the Blue Whiting. Overall, I find this paper novel, insightful and very valuable. The topic this paper addresses is extremely relevant and timely. The paper is structured appropriately and overall well-written.

I have just two general suggestions for the authors to consider. I then present a few more specific comments indicated by the line number. Overall, I suggest publication after addressing those comments deemed relevant below.

We thank the reviewer for their kind comments and enthusiasm for our work. We have addressed the comments below.

General suggestions

First, the authors indicate starting from line 216 that they use models to predict changes in habitat, not distribution, and clearly explain the important difference between the two concepts. I suggest that this caveat is moved to the front of the paper, and that the remaining text is adjusted to reflect that the authors are not attempting to model distribution, but habitat. Now there are several lines in the paper that would suggest otherwise (including the title), and I suggest adjusting the language throughout. Some examples: lines 34, 76 (remove distribution), line 107 (are these not primarily 'habitat drivers' then as there are many more distribution drivers not included here as explained by the authors?), lines 227, 252 etc.

However, if there is some overlap between habitat and distribution modelling (which there seems to be), I suggest explaining more clearly where the models used by the authors situate themselves in that space.

While “habitat” and “distribution” are often seen as interchangeable, we have tried here to be very clear and consistent about their use. However, in some cases the ability to predict habitat allows us to make inferences about the actual distribution that will be observed: this is the point that we make in the paragraph starting from line 216 (in the original Word submission). We agree however that it is a good idea to introduce this distinction earlier in the manuscript and have added the following text to the paragraph starting on line 76 (in the original Word submission):

Here we demonstrate the ability to forecast shifts in the habitat and distributions of marine species on the decadal scale as one of the first examples of decadal climate predictions directly applicable in a decision-making context. *We first draw a necessary distinction throughout this work between the habitat of a species (where conditions are suitable for occurrence) and its distribution (where it is actually found).* We focus on three exemplar fish species....

We have checked the usage of habitat and distribution throughout the manuscript again and feel that they are appropriate and reflect this distinction, with the exception of the caption of Figure 1 (line 107 in original Word submission), which we have corrected.

Second, I would suggest a bit more clarity on why the following elements were selected:

a) Why this region?

b) Why these species?

c) Why salinity and SST for blue whiting and mackerel/bluefin respectively?

d) Why the particular months (i.e. August and March)?

e) Why this particular region for the mackerel?

I can see that much, if not all, of this information is located in the supplementary material, but just a brief justification would be useful for the reader. For a), it seems that this region was most predictable and therefore a most useful case for your modelling purposes (from lines 368-369), but this would potentially be more difficult in other regions? I suggest briefly discussing this (i.e., the applicability of this method in data-poor regions or less predictable areas). For b), these species were likely chosen primarily for their (large) shifts in distribution? I suggest clarifying that. For c) and d), I can see that this is linked to a literature review on the most important habitat drivers for the species. I would suggest just stating this briefly (e.g., in line 98: “from our review/ from recent findings, this is the primary environmental driver...”), so the non-expert reader knows why you have made this choice. Lastly, for e), I am wondering why the authors decided to focus on the area around Greenland (from Fig1), not Iceland, as it is the Icelandic participation in the mackerel fishery (in its own EEZ) that has sparked the conflict with the EU, Norway and Faroe Islands.

While the reviewer has correctly answered all of these questions themselves, it is also clear that we need to be more explicit about the reasoning behind our choices. We have therefore tweaked the section of the Methods entitled “Fish Species and Habitat Models” so that it is now an explicit discussion of these choices, and answers to all of the reviewers questions can now be clearly found here. We have also taken on the reviewer’s suggested approach to answering c and d in the main text so that it is clear that the choice of variables is linked back to the literature on the topic (and to Extended Data Table 1). The point around applicability to other regions has already been addressed in the last paragraph of the main text.

The choice of Greenland vs Iceland was driven primarily by the availability of the necessary habitat models. Greenland appears to be at the furthest edge of the NE Atlantic mackerel distribution and the temperature

limitation on mackerel distribution appears to be strongest here (particularly in the waters around the cold East-Greenland current), making it suited to environmentally-driven predictions. Jansen et al 2016 in particular studied the processes limiting mackerel distribution here and used the knowledge to make projections under a changing climate. While temperature has been highlighted as playing a role in shaping the distribution around Iceland as well in other studies (e.g. Nikolioudakis et al 2019, Boyd et al 2020), other factors that are not available from the climate prediction models (e.g. phytoplankton, zooplankton and herring abundances) were also important in these models. Furthermore, these distribution models are not available in the public domain and their high complexity (and use of closed data sources for parameterisation) means that they cannot readily be reproduced. We therefore chose to perform our analysis parallel to the Jansen et al paper, focussing on the simpler case study around Greenland where habitat is primarily limited by temperature. We have added text to the section on mackerel in “Fish Species and Habitat Models” to reflect this rationale more explicitly.

Specific comments

I.92: Is it possible for the authors to indicate for how many years/species they have observations of distribution?

There is good evidence of distribution shifts for all three species – this is indeed why we chose them as case studies in the first place. For blue whiting we have a time series from a scientific survey (Figure 4b) that is designed to cover the full distribution of the spawning population back to the early 2000s, and therefore captures the expansion and contraction equally. However, the observations of mackerel and Bluefin tuna distribution shifts are more opportunistic in nature, and these stocks have not been the subject of rigorous scientific monitoring efforts in the same manner – it is therefore not possible to come directly with a time series of distribution that we could compare against. We have clarified the nature of the observational data in the materials and methods section (see section on “Distribution and Abundance Data”) but due to their relatively detailed nature we have chosen not to make a change to the main text as the reviewer suggested.

I.212: I am wondering about the reliability of scientific monitoring surveys to estimate distributional area, as often areas where species could be are not monitored if it is a new distribution pattern. Could this throw off some of the models’ predictive capacity?

This is a valid point and is a general and recurring issue in the design of fisheries monitoring surveys. The primary purpose of data collected by the International Blue Whiting Spawning Stock Survey is to estimate the abundance of this species, a key input in the species’ assessment and management. The survey is a large undertaking involving 4-6 research vessels from European nations over several weeks and employs an adaptive design to ensure full coverage of the stock: close collaboration with commercial fishing vessels also helps guarantee that all fish have been counted. The data used is therefore the best available but a risk still remains that part of the distribution has been missed in a given year. However, as we consider performance over multiple years, and each distribution and survey is largely independent of previous ones, it is unlikely that there is a systematic underestimation of the distribution (even if it may occur in a single year). We therefore see such an issue to be unlikely in our analysis.

I.240: Could the authors explain ‘forecast initializations’ briefly for the non-experts?

We have refocussed this paragraph and moved it earlier in the manuscript to make the rationale for a comparison against uninitialised forecasts clearer, following a suggestion from Reviewer 2: in doing so, we have also included a clarification about initialisation and hope that this is clearer now.

We thank the reviewer once again for their constructive and very useful comments.

Reviewer #2 (Remarks to the Author):

I was extremely excited about the invitation to review this manuscript. The field of ecological forecasting is in its infancy, particularly in the marine domain, and this manuscript is the first (that I know of) to show skillful decadal-scale predictions of ecological metrics. This multi-annual time horizon is a critical missing gap in ocean predictability yet it is the time horizon most often sought after by decision makers. This manuscript is a huge scientific advance, shows novel results, and I anticipate will be a pioneering manuscript for the field of ecological forecasting. I congratulate the authors on a well-written and exciting manuscript.

In summary, the manuscript shows decadal predictability of habitat for three marine species of fisheries interest, mackerel, bluefin tuna, and blue whiting. The authors use output from the CMIP6 decadal prediction system as the forcing for their retrospective forecasts, and discuss the implications of an operational product for fisheries stakeholders. Overall, I thought the manuscript was well written, the methods sound, and the results offered a huge advancement in our understanding of ecological predictability. I have some minor points below that I think, if addressed, could be helpful to improve clarity for a future reader.

We thank the reviewer for their kind comments and are pleased that they share our enthusiasm for this work.

First, I didn't fully appreciate the value of the prediction (initialized forecasts) versus projection (uninitialized forecasts) comparison. The primary text (In 239-250) is fine but it doesn't really consider that predictions and projections offer vastly different outlooks by design, the latter being a scenario planning exercise. I can't imagine anyone using a projection as a prediction but perhaps I misunderstand. I feel like this part of the paper (i.e. In 239-250 + extended figure) detracts from the main message of skillful decadal predictions. If you need additional space (pending reviews) then I would opt to remove this section, but I appreciate you may disagree with my perspective. If it stays in then I suggest adding a sentence or two better describing the need for this comparison.

Our rationale for including the comparison with uninitialized models was to address the question “does initialisation improve the forecasts?”: this type of comparison is very common in the climate prediction community for example. The use of “projections as predictions” is perhaps not such a crazy idea, particularly given that the external forcing scenarios used in projections are essentially indistinguishable from each other in the near term. One could imagine a situation where initialisation gives very little added value and all of the skill arises from the external forcing (this is actually the case in some parts of the global ocean). In such a case, the use of projections would certainly be justified, particularly as it both substantially reduces the amount of data to be processed and simplifies the analysis procedure (particularly with bias correction). We therefore viewed it as necessary to demonstrate that the use of initialised forecasts (instead of just uninitialized projections) is warranted and does improve our habitat forecasts. We have made this point more clearly in the revised text by moving the paragraph in question earlier in the manuscript, and expanding the text to elaborate the point.

Second, it wasn't until I reached the methods section that it became apparent that the 'habitat models' for bluefin and mackerel were not an ecological model, but rather the area of a thermal preference. I don't have an issue with the metric itself, but I think it could be made clearer in the main text (perhaps In 95-97) that you are using an isotherm threshold to identify species habitat, which is distinct from the blue whiting approach that uses a habitat model. To me this is an important distinction because physical-only forecasts are inherently more skillful as they don't integrate the uncertainty associated with an ecological model.

We also see “thermal preference” as a form of ecological model – it is after all a simplification and parameterisation of the way that species interact with their environment. We are however a little concerned that starting to distinguish between typologies of models may confuse the reader more than it benefits them, and therefore have chosen to leave the main text as it is. We have however revised Extended Data Table 1 to reflect the reviewer’s suggestion and describe the habitat models used more clearly. We hope that this is satisfactory.

Third, I really valued the integration and comparison with a persistence forecast. While I think comparing with persistence is important, I’m under the impression it’s also general practice to detrend results to separate skill from the climatology. I.e. how well do the models perform at predicting anomalies rather than actual SST. It seems like MESS might be doing this (I’m not familiar with this metric) so perhaps no additional work is needed, but it might be useful to add a sentence somewhere indicating that you have skill above the climatology and above persistence (say somewhere in the paragraph from Ln 126-141).

Skill scores based on detrended values are indeed used in some fields, again particularly in climate prediction where there is a desire to separate the skill into its various components (e.g. to separate internal variability from external forcing). Here we have tried to quantify the skill from the perspective of the potential user and decision maker (who need skilful forecasts), rather than from the view of scientists (who need to understand their forecast system). We have therefore worked directly with non-detrended indicators.

We also note that because we have bias corrected all of our physical variables, skill metrics between anomalies will be identical to those between the raw values (as the forecast and observed values share a common climatology). We have not made any changes in response to this comment.

Finally, I had some trouble understanding the Pearson correlation significance threshold that you used (e.g. Figure 1). Typically, I’ve seen significance reported from the Pearson correlations itself, not whether the correlation is great than 0. For example, for 59 years of SST an r of ≥ 0.26 is significant; for 34 years of salinity a r of ≥ 0.34 is significant. Could you perhaps explain your approach more fully in the methods somewhere?

We describe the approach to calculating the significance of all skill metrics in the methods (under “Forecast Verification and Skill”). The approach taken here is a bootstrapping one, based on resampling (with replacement) the data used to calculate the skill metric. An alternative approach would be to use a parametric test, such as the Fisher r -to- z transformation, but regardless, both approaches generate a distribution of values around the central estimate (from which a confidence interval or significance threshold can be calculated). We chose to interpret this as a one-tailed test (i.e. significantly greater than zero) as we feel that this is the key question that users would ask – is the forecast skill significantly *better* than a given threshold for uptake (e.g. zero, persistence, uninitialised projections etc). At the end of the day, the approach is very similar to the one the reviewer describes: rather than calculating the threshold for a given significance level (e.g. 95%), we have calculated the p value for a given threshold (i.e. correlation > 0). We have expanded the relevant section in the methods a little to elaborate on our calculation of significance levels.

Line-specific comments:

- Ln 30-35: This is fine but I actually think it undersells the advancement you offer to ecological forecasting. You don’t have to make any changes, but I do think you could take the opportunity to more strongly spell out that (1) decadal ecological forecasts haven’t been shown before; (2) that multi-annual forecasting is highly desired by end-users, and more so than at any other lead time; (3) that there is a novel research gap that needs to be filled for this specific forecast horizon.

We appreciate the reviewers enthusiasm for our work, and have tried to make these points clearer throughout the text, without overselling the work.

- Ln 76-78: *I thought this sentence could be made a bit more punchy and less verbose. For example: "Here we demonstrate one of the first examples of a decadal forecast of shifts in the habitat and distributions of marine species"*

Thanks for the tip. We have honed the sentence along the lines suggested.

- Ln 98-99: *'five-year ahead predictive skill' is a bit awkward. Why not say 5-year forecast skill? Or use 'lead time' terminology?*

Ugh – apologies for the horrible phrasing! We have tweaked the sentence!

- Ln 118: *add units here for the area of suitable habitat*

Done, with additional tweaks to this paragraph to make it less verbose.

- 165-179: *great paragraph! I really like how you identified the mechanism of predictability here. If useful, Brodie et al. 2021 also showed that species distributions are more predictable at these low frequency scales as well.*

Thanks for the tip on the very relevant reference - we have included it.

- *Methods: Great – well detailed and laid out nicely!*

Brodie, S., Abrahms, B., Bograd, S. J., Carroll, G., Hazen, E. L., Muhling, B. A., ... & Jacox, M. G. (2021). Exploring timescales of predictability in species distributions. Ecography. <https://doi.org/10.1111/ecog.05504>

I look forward to seeing this published.

Stephanie Brodie

Thanks once again for the kind and very constructive comments.

Reviewer #3 (Remarks to the Author):

This paper describes an assessment of skill for multiyear predictions of monthly averaged ocean properties within domains relevant to fish habitat, and hence fisheries, of the North Atlantic. The authors use a "grand ensemble" of model results for their skill assessment, and the size of that ensemble is large enough to be statistically meaningful. They include a comparison of model-based forecasts with persistence forecasts, to demonstrate the added value of the model results. Their focus on the North Atlantic is sensible, as decadal forecasting results are especially skillful in that region (such skill varies widely over the world ocean). I appreciated the discussion of potential advantages/pitfalls of using these forecasts in real-world fisheries management. I also liked the declarative style of the captions.

I was very much impressed by the writing and presentation of this work. I have no objection to publication of the manuscript in its present form. I think this is a valuable contribution which will encourage stakeholders to utilize these emerging predictive tools, as they demonstrate some genuine skill. I do have a few comments and suggestions, as follows:

Fig. 1. Can the authors shed any light on why the pattern of skill looks as it does? In particular, is there some correlation between skill and bathymetric depth? They do note some of the general processes leading to skill in the paragraph beginning at line 168, but the reader may wonder what factors lead to enhanced/degraded skill on some areas, as opposed to it just being a random artifact of a necessarily limited (although reasonably large) ensemble size.

This very good question unfortunately does not have an easy answer. Many factors determine the spatial skill pattern and this is one of the major themes addressed by the field of climate prediction. A few points that we would highlight in particular are 1) the inherent ability of each model system to represent key processes, particularly in regions of complex bathymetry and 2) the availability of data to initialise the system: as we note in the manuscript, we discovered clear issues in the sub-surface salinity data prior to 1985. Figure 1 represents the (net) sum of these and other processes across the five different model systems that we consider here, and a direct interpretation is unfortunately both very challenging and beyond the scope of this manuscript. We have therefore not gone into detail about the origins of these patterns: however, we have highlighted more clearly relevant literature reviews that we refer to are particularly useful and accessible in this regard if the reader or reviewer wishes to delve deeper.

I. 137 "skilful" should be "skillful"

We have used British English spelling, where "skilful" is the norm, throughout our manuscript (and including the title).

I. 203 I agree that such forecasts would have been useful had they existed in 1990, but of course the technology (modeling capability) was not available at that time. But no objection to using this as an example.

☺ Yes, we have added "had they been available" to make this point a bit clearer!

I. 243 and I. 465 Predictions vs projections - It seems obvious to this reader that a forecast initialized with observed/reanalyzed ocean conditions will do vastly better than one initialized with an alternate realization of the "present" ocean, and that one would expect that advantage to decrease over time as the initial conditions are forgotten by the system. Apparently this type of comparison ("initialized" vs. "uninitialized") is standard in the field (e.g. as described in the Yeager et al. 2019), and it is potentially useful to see it quantified. I would prefer a bit more detail about what forcing applies to both runs (presumably, greenhouse gases and variations in the solar constant), what biases might be shared by both runs (e.g. climatological drift), and how the "uninitialized" runs themselves were spun up. As it stands, I needed to consult with Yeager et al. reference for these important details.

All three reviewers have commented on this paragraph! We have combined this reviewers comments with those of the other reviewers comments to improve this paragraph and hopefully it is more accessible to those not directly involved in the decadal prediction community now. The methods section already provides details about the forcings used, under "Decadal Forecast Models" and "Unitialised projections". Many of these issues are standard knowledge in climate prediction and modelling community and so we have not gone into the details here: we have however modified these two sections to point the reader to both key technical references and more articles providing general background. We hope that this is sufficient.

*l. 489 (Bias correction section) - I assume the climatological field of each variable calculated as a function of model and forecast lead time *and* start date (day of year) of the forecast. If so, please state explicitly.*

This would be appropriate in the case where there were multiple start dates per year (as is typically seen in e.g. seasonal forecast systems). However decadal predictions are typically only initialised once per year, and therefore we do not need to correct for start date in this case. We have added a sentence to the “Bias correction” section of the methods to clarify this point

l.512 "hindcasting" - many oceanographers use this term to mean atmospherically-forced simulations of past ocean states, so I am glad to see the authors avoid using that term for the retrospective forecasts.

Yes, exactly! This particular point caused many headaches when first starting out in the field, which is why that sentence was included!

In conclusion we thank the reviewer for their kind and supportive comments.

*** See Nature Research's author and referees' website at www.nature.com/authors for information about policies, services and author benefits.*

This email has been sent through the Springer Nature Tracking System NY-610A-NPG&MTS

Confidentiality Statement:

This e-mail is confidential and subject to copyright. Any unauthorised use or disclosure of its contents is prohibited. If you have received this email in error please notify our Manuscript Tracking System Helpdesk team at <http://platformsupport.nature.com> .

Details of the confidentiality and pre-publicity policy may be found here

<http://www.nature.com/authors/policies/confidentiality.html>

Privacy Policy | Update Profile

DISCLAIMER: This e-mail is confidential and should not be used by anyone who is not the original intended recipient. If you have received this e-mail in error please inform the sender and delete it from your mailbox or any other storage mechanism. Springer Nature Limited does not accept liability for any statements made which are clearly the sender's own and not expressly made on behalf of Springer Nature Ltd or one of their agents.

Please note that Springer Nature Limited and their agents and affiliates do not accept any responsibility for viruses or malware that may be contained in this e-mail or its attachments and it is your responsibility to scan the e-mail and attachments (if any).

Springer Nature Ltd. Registered office: The Campus, 4 Crinan Street, London, N1 9XW. Registered Number: 00785998 England.

REVIEWERS' COMMENTS

Reviewer #2 (Remarks to the Author):

The authors have addressed my concerns. I look forward to seeing this work published.

Reviewer #3 (Remarks to the Author):

I have now examined the revised version of the Payne et al. paper entitled "Skillful decadal-scale prediction of fish habitat and distribution shifts". The authors appear to have made the improvements requested by the reviewers. In particular, their further clarification of "uninitialized" vs "initialized" forecasts (in the paragraph beginning on line 161 of the pdf showing revisions) is appreciated by this reviewer. I would suggest a slight grammatical correction, i.e.

l. 161 change "habitat forecasting to could to be use" -> "habitat forecasting could be to use"

other than this small grammatical correction, I have no objection to publication of the revised manuscript in its present form.

-Albert J. Hermann

Reply to Editor and Reviewers

In the following point-by-point response we have included the editors / reviews comments in blue italics, and our reply in black, normal type.

Reviewer #2 (Remarks to the Author):

The authors have addressed my concerns. I look forward to seeing this work published.

Reviewer #3 (Remarks to the Author):

I have now examined the revised version of the Payne et al. paper entitled "Skilful decadal-scale prediction of fish habitat and distribution shifts". The authors appear to have made the improvements requested by the reviewers. In particular, their further clarification of "uninitialized" vs "initialized" forecasts (in the paragraph beginning on line 161 of the pdf showing revisions) is appreciated by this reviewer. I would suggest a slight grammatical correction, i.e.

l. 161 change "habitat forecasting to could to be use" -> "habitat forecasting could be to use"

other than this small grammatical correction, I have no objection to publication of the revised manuscript in its present form.

-Albert J. Hermann

We have made the change suggested by Reviewer 3. We would also like to thank the reviewers for their contribution one more time and are pleased that our changes are satisfactory.